# The Effects of Staining and Cleaning on the Color and Light Transmittance Changes of a Copolyester Retainer Material with Different Surface Textures

**DOI:** 10.3390/ma15196808

**Published:** 2022-09-30

**Authors:** Laurie Susarchick, Insia Virji, Grace Viana, Mervat Mahmoud, Veerasathpurush Allareddy, Max Gruber, Henry Lukic, Spiro Megremis, Phimon Atsawasuwan

**Affiliations:** 1Department of Orthodontics, University of Illinois Chicago, Chicago, IL 60612, USA; 2Dental Materials & Devices Research—American Dental Association, Chicago, IL 60610, USA

**Keywords:** orthodontic, clear retainers, copolyester, light transmittance, color stability, color change

## Abstract

This study evaluated the efficacy of different cleaning methods with respect to changes in the color and light transmittance of both rough and smooth thermoformed, copolyester retainer specimens, after staining in different solutions. Four hundred copolyester sheets (Essix ACE) specimens were fabricated over molds with different surface textures, smooth and rough. The specimens were stained in four different solutions (*n* = 100 per solution) over 28 days; then, each of the four groups of 100 stained specimens was sub-divided into five groups of 20 specimens and subjected to a different destaining solution. The specimens were sub-divided with half subjected to an additional ultrasonic cleaning procedure. Light transmittance and color changes were analyzed using a spectrometer/integrating sphere assembly and a spectrophotometer. Mean difference comparisons were performed using appropriate statistical tests at *p* = 0.05. All five destaining solutions proved to be effective at removing coffee and tea stains. The surface roughness of the retainer material plays a significant role in the ability of cleaning solutions to remove stains, demonstrating a greater effect on cleaning rough specimens with respect to improvements in light transmittance and greater changes in color. Additionally, an ultrasonic cleaning unit generally enhanced the ability of all five solutions to clean the tea-stained specimens. However, the enhancements were only significant for light transmittance.

## 1. Introduction

In orthodontics, one of the most common concerns for patients and providers is post-treatment relapse, which can lead to both time and economic burdens. Relapse occurs when teeth return to their original position after orthodontic treatment. Retainers are the only effective approach to prevent relapse [1]. Clear retainers have gained in popularity due to their aesthetic, clear appearance [2,3,4]. However, to maintain this clear appearance over long time periods, effective cleaning of the retainer is necessary. The transparency and color stability of clear retainers remain critical considerations for both patients and clinicians [5]. To date, only a few studies have investigated the long-term effectiveness of cleaning solutions for maintaining the color stability and light transmittance of clear thermoplastic retainers [6,7,8]. Yet, the characterization of the ability to remove “stains”, once they have occurred, requires further investigation [9]. 

Generally, thermoplastic clear retainers can be fabricated by thermoforming plastic sheets over either a plaster dental model or a 3D-printed dental model. These models possess different surface textures (roughnesses), which are then transferred to the internal surfaces of the thermoplastic clear retainers [10,11]. To our knowledge, there is no study that investigates the effects of different surface textures on changes in color and light transmittance of clear retainer materials that have been stained and then cleaned using different cleaning methods. Therefore, it is a goal of this study to establish an evidence-based method for the cleaning of a clear retainer material with both a clinically relevant, rough surface and a smooth surface by investigating the ability of different cleaning methods to make changes in light transmittance and color of specimens after they have been stained in various solutions.

## 2. Materials and Methods

Four-hundred specimens were fabricated by thermoforming sheets of copolyester material [12] (Essix ACE^®^, Dentsply Sirona, Inc., Charlotte, NC, USA) over aluminum molds, as shown in Figure 1A,B. All sheets were 1 mm (0.040″) thick and thermoformed using the Biostar^®^ Scan pressure machine (Scheu-Dental), according to the manufacturers’ instructions. Three individual specimens were machined from each of the thermoformed sheets with nominal dimensions of approximately 50.8 mm long by 12.7 mm wide by the final thermoformed thickness (less than 1 mm).The rectangular shapes on the aluminum molds had pockets for textured inserts, which imparted a clinically relevant average surface roughness on approximately half of the length of the resulting specimens, as shown in Figure 1A.The inserts were acrylonitrile butadiene styrene (ABS) printed using a Mojo 3D Printer version 3.1 (Stratasys Inc, Eden Prairie, MN, USA). The average area roughness parameter, Sa, of the inserts was ~10.5 µm with a coefficient of variation (c.o.v.) of 3%, which was measured using a Zygo New View 8300 optical interferometer. This Sa value was in the range of values obtained by measuring Invisalign retainers from the clinic: overall mean Sa = 10.6 µm with a standard deviation (s.d.) of 9.1 µm and c.o.v. of 86%; buccal mean Sa = 17.5 µm with a s.d. of 13.7 µm and c.o.v. of 78%; and lingual mean Sa = 17.5 µm with a s.d. of 13.7 µm and c.o.v. of 78%.

The four hundred specimens of Essix ACE^®^ material were divided into four groups of 100 specimens in each group exposed to a different staining solution, as shown in Table 1. The chosen staining solutions were distilled water (control), instant coffee, black tea, and red wine, based on their capability to stain clear retainers in the published literature [9,13,14].

The instant coffee solution was prepared by mixing 688 g of instant coffee powder (Nescafe^®^ Original, Nestle Ltd., Vevey, Switzerland) into 8 L of distilled water (ISO grade 3), as per the manufacturer’s instructions [14]. The black tea solution was prepared by mixing 150 g of instant tea powder (Lipton^®^ Unsweetened Black Tea Mix, Nestle Ltd., Vevey, Switzerland) into 8 L of distilled water (ISO grade 3), according to the manufacturer’s instructions. The red wine used was a Cabernet Sauvignon (Paint Box^®^ Cabernet). Distilled water (ISO grade 3) was used as a control solution. 

The specimens were immersed in the freshly made staining solutions and held at 37 °C in an incubator for 28 days. Color and percent light transmittance measurement were taken at day 0 (baseline) and then at days 7, 14, and 28, as described below. After staining for 28 days, each group of 100 specimens was divided into five smaller subgroups of 20 specimens each for the destaining experiments. The specimens were then subjected to five cleaning solutions: Invisalign^®^ Cleaning Crystals (sodium sulfate-based), Retainer Brite^®^ (sodium perborate-based), Polident^®^ (sodium bicarbonate-based), Listerine^®^ mouthwash (21.6% ethanol), and 3% hydrogen peroxide (H_2_O_2_). Furthermore, for each cleaning solution, half of the 20 specimens were cleaned in an ultrasonic cleaner unit (BioSonic^®^ UC300R, Coltene/Whaledent, Altstätten, Switzerland) at a frequency of 42,000 Hz for 15 min, while the other half of the specimens were subjected to the cleaning solutions without ultrasonication using stirring with a magnetic stirrer (Figure 2) [6,7,8]. The stained specimens were destained in the cleaning solutions in groups of 10 for 15 min each, except for the specimens in the Polident solution, which were soaked for 3 min, as specified in the manufacturers’ instructions. The specimens were kept in artificial saliva [15] at 37 °C between measurements. The breakdown of staining and destaining groups is shown in Table 1.

Absolute percent light transmittance was determined based on a previously published method for measuring the translucency of dental ceramics due to the standardized specimen preparation and well-established method [16]. Briefly, this method quantifies the percent of light transmittance through the retainer material into a spectrometer/integrating sphere system consisting of the following components: a miniature spectrometer (Flame-S-VIS-NIR, Ocean Optics Inc., Dunedin, FL, USA); an X-Cite^®^ 120 Fluorescence Lamp Illuminator and liquid light guide (Excelitas Technologies Corp, Waltham, MA, USA); an integrating sphere (Labsphere Inc., North Sutton, NH, USA); a fiber optic cable (QP100-2-UV-VIS, Ocean Optics Inc., Dunedin, FL, USA); and a custom-designed specimen holder. During the measurement procedure, a light energy reading is taken with the light source connected to the spectrometer/integrating sphere system through a custom-fabricated specimen holder attached to a port in the integrating sphere, as illustrated in Figure 2. Next, a specimen is positioned in the holder in the path of the light source, and the light energy reading is taken with the light transmitted through the specimen (Figure 2). From the two light energy measurements, the percent of light transmittance through the specimen is calculated for wavelengths between 380 nm to 780 nm (Oceanview software, version 1.5, Ocean Optics, Dunedin, FL, USA). Changes in percent light transmittance, ∆T, were calculated for days 7, 14, and 28 with respect to baseline. Changes in percent light transmittance, ∆T, were also calculated following destaining using the different cleaning methods, comparing light transmittance values at day 28 of staining with values measured after the respective cleaning methods.

The color of the specimens was evaluated with respect to the International Commission on Illumination L*a*b* (CIELAB) color space [17] using a Konica Minolta CM-2600d Spectrophotometer (Figure 2c). In this color space, L* [18] is a measure of lightness from 0 (black) to 100 (white) [5], and a* and b* are chromaticity coordinates: +a* is red direction; −a* is green direction; +b* is yellow direction; and −b* is blue direction. At each measurement time point, each specimen was measured in triplicate at the same position on the specimen, and average values were calculated automatically and recorded. Using the CIELAB color space, color difference can be expressed as a single value, using the following equation [18]: ∆E* = [(∆L*)^2^ + (∆a*)^2^ + (∆b*)^2^]^½^(1)

For each specimen, color changes were determined by using the recorded average CIELAB values and calculating differences in ∆E* values with respect to time points, surface textures, staining treatments, and cleaning methods. All measurements were conducted in the same room with standardized lighting. The national bureau of standards (NBS) system was then used to describe visual color change in terms of NBS values, as follows [16]:NBS = [(∆E* at specific day of measurement − ∆E* at measurement baseline) × 0.92](2)

NBS values above 3.0 were considered to have a marked change in color [9], which for this study was considered clinically unacceptable (Table 2).

Based on Shapiro–Wilk normality test, parametric or non-parametric tests were appropriately used for testing group differences. Additionally, the following statistical analyses were performed, as appropriate: independent Student’s *t*-tests with the Levene’s test for equality of variances; one-way ANOVA with a test for homogeneity of variance; Kruskal–Wallis; and Mann–Whitney using the Bonferroni correction. Statistical significance was set at 5%. IBM SPSS statistics for Windows, version 28.0 (IBM Corp: Armonk, NY, USA) was the software used for the statistical analyses.

## 3. Results

Table 3 shows the descriptive and statistical analyses of the changes in percent light transmittance, ∆T, for the specimens with different surface textures at different times for the three staining solutions and the control. Importantly, Table 3 shows that both the rough and smooth specimens were significantly stained by the coffee, tea, and wine solutions with respect to ∆T values at day 28 compared to the values for the respective specimens at days 14 and 7. The only exception was that the ∆T values for the rough specimens stained in wine at day 28 were greater than the values for the rough specimens at day 14, but the difference was not significant. Furthermore, by day 28, the coffee and tea staining solutions had a more dramatic effect on ∆T values compared to the staining solution of wine and the control of water. That is, at day 28, the ∆T values for both the rough and smooth specimens in coffee and tea were greater than the values for both the rough and smooth specimens in wine and water. This is also shown qualitatively in Figure 3. Most importantly, it was shown that after 28 days, the coffee and tea staining solutions proved to be especially effective at decreasing the percent light transmittance through the specimens, so that the effect of the cleaning solutions could be evaluated with respect to this parameter for the specimens stained in those solutions.

Likewise, by day 28, the coffee and tea staining solutions had a more dramatic effect on color change compared to the staining solution of wine and the water control. The details of this effect are shown in Table 4. Importantly, Table 4 shows that both the rough and smooth specimens were significantly stained by the coffee and tea solutions with respect to NBS values at day 28 compared to the values for the respective specimens at days 14. Additionally, again, at day 28, the NBS values for both the rough and smooth specimens in coffee and tea were greater than the values for both the rough and smooth specimens in wine and water. Similar to the ∆T values, wine did stain the specimens with respect to color change; however, the results were not always significant. Additionally, by day 28, all of the specimens stained in coffee and tea had mean NBS values that were 12 or above, indicating “very much change to other color”, while the average NBS values for the specimens stained in wine were only “noticeable” at 3 or less (Table 2). Note that, by day 28, surface texture did not have a significant effect on the ability of the staining solutions to stain the respective specimens soaked in them with regard to ∆T and NBS values.

After 28 days of staining, each group of stained specimens was subjected to the different destaining processes. However, after 28 days, since only the specimens stained in the coffee and tea solutions showed significant staining with respect to changes in both ∆T and NBS values compared to the control specimens in water, only the results and analyses for the cleaning methods on the coffee- and tea-stained specimens are emphasized in the study.

For the specimens stained in the coffee solution, the surface roughness of the specimens had a statistically significant effect on destaining with respect to both ∆T and NBS values, as shown in Table 5 and Table 6. For all five destaining solutions, these tables show that the rough specimens were cleaned to a significantly greater extent than the smooth specimens with respect to greater changes in both ∆T and NBS values. This same trend was shown for the destaining of the rough and smooth specimens that had been stained in tea; however, the results for the tea-stained specimens were only statistically significant for the NBS values and not for the ∆T values.

Specifically, for the rough specimens stained with coffee, all five destaining solutions resulted in changes in percent light transmittance values of about 22%, with only the Retainer Brite solution demonstrating a significantly larger ∆T compared to the rough specimens destained with Listerine (Table 5). Additionally, for the smooth specimens stained in coffee, all five destaining solutions resulted in ∆T values of about 16%, with the Invisalign Crystals solution demonstrating a significantly larger ∆T compared to the rough specimens destained with all the other destaining solutions except Retainer Brite. Similarly, for the specimens stained with tea, all five destaining solutions resulted in changes in percent light transmittance with the ∆T values for rough specimens improving by about 24% and for the smooth specimens by about 21% (Table 5). However, the only statistically significant differences were found among the rough specimens, with the H_2_O_2_ solution being significantly less effective than the other solutions except for the Polident solution.

Similar trends were shown for the results of the color change in the specimens after application of the destaining solutions. Specifically, for the rough specimens stained with coffee, all five destaining solutions demonstrated NBS values of about 15, except for the specimens destained in H_2_O_2_ with an NBS value of about 14, which was a significantly lower effect than for the specimens destained in the Invisalign Crystals and Retainer Brite solutions (Table 6). Likewise, for the smooth specimens stained with coffee, all five destaining solutions resulted in NBS values of about 11 with the specimens destained in H_2_O_2_ having a significantly lower color change compared to those in the Invisalign Crystals solution. Furthermore, for the rough specimens stained with tea, all five destaining solutions demonstrated NBS values of about 15 with the specimens destained in H_2_O_2_ having a significantly lower change compared to the specimens in all of the other destaining solutions, except for those in the Listerine solution (Table 6). Similarly, for the smooth specimens stained with tea, all five destaining specimens demonstrated NBS values of about 13 with the specimens destained in H_2_O_2_, again, having a significantly lower change compared to the specimens in all of the other destaining solutions, except for those in the Listerine solution.

As for the effect of an ultrasonic cleaning unit on enhancing the ability of the different destaining solutions to clean the specimens, those stained with the coffee solution showed no statistically significant effect from the ultrasonic cleaning unit for either ∆T or NBS values for any of the destaining solutions (Table 7 and Table 8). Yet, for the specimens stained in the tea solution, an ultrasonic cleaning unit generally enhanced the ability of the destaining solutions to clean the specimens with respect to both ∆T and NBS values; however, the results were only significant for the ∆T values. The specimens destained in the different solutions with the use of an ultrasonic cleaning unit had an average ∆T of about 24 to 25, and those without the use of an ultrasonic cleaning unit had average ∆T of about 21 (Table 7).

## 4. Discussion

It is said that the only effective approach to prevent orthodontic relapse and achieve a stable result is long-term retainer wear [20]. With the advancements of orthodontic techniques, clear retainers have increased in popularity due to their aesthetic nature and comparable treatment times [1]. Since clear retainer material interactions with the oral environment can occur, resulting in plaque and calculus buildup along with bacteria buildup and retention, clear retainers must be maintained to mitigate loss of their light transmittance and maintain material integrity [5,6,7,8,21]. Color stability can also be affected by ultraviolet radiation, mouthwash, and various beverages [5,6,7,8,22]. Additionally, in the United States, 50% of Americans over 18 years old drink coffee, and coffee drinkers consume an average of three cups of coffee per day [23]. Providers should clearly inform patients to remove retainers before eating and drinking, as staining could occur from food or drinks that would allow the stain to accumulate on the retainer [9].

It has been suggested that different thermoplastic materials react differently when exposed to staining and destaining solutions [6,7,8,9]. For instance, the acidic nature of wine and coffee can cause surface roughening conducive to staining. In particular, the tannic acid found in coffee has been reported to be responsible for the yellow-brown color associated with both the absorption and adsorption of the ingredient [24]. Furthermore, red wine has been reported to cause severe staining on provisional resin materials [25,26]. Additionally, Bernard et al. found that tea caused discernible extrinsic stains on the surfaces of aligners, but they were easily removed [9].

In this study, we evaluated the effect of surface textures on the staining and destaining effects of the materials. In addition, the staining and destaining effects were evaluated on the same sets of the specimens, which provided comprehensive information on the light transmittance and color stability of the copolyester specimens. Previous research has shown a copolyester retainer material exhibited greater color change as wear time increased with coffee being the most prominent staining agent followed by tea and red wine [21]. With respect to color changes, our study showed similar staining results. After 28 days in the staining solutions of coffee, tea, and wine, the NBS values for both the rough and smooth specimens in coffee and tea were significantly greater than the values for both the rough and smooth specimens in wine and water. Furthermore, the specimens stained in coffee and tea had mean NBS values that were 12 or above, indicating “very much change to other color”, while the average NBS values for the specimens stained in wine were only “noticeable” at 3 or less. Additionally, by day 28, the ∆T values for both the rough and smooth specimens in coffee and tea were significantly greater than the values for both the rough and smooth specimens in wine and water. Since only the specimens stained in the coffee and tea solutions showed significant staining with respect to changes in both ∆T and NBS values compared to the control specimens in water, only the results for the cleaning methods performed on the coffee- and tea-stained specimens were further analyzed for effectiveness.

The selected cleaning solutions in this study were chosen based on their performance in previous studies [6,7,8]; these studies showed that, after 6 months of exposure to the cleaning solutions, specimens exhibited the least amount of change in percent light transmittance values compared to baseline values. Furthermore, Invisalign Cleaning Crystals, Retainer Brite, and Polident are all widely available and commonly used to clean orthodontic retainers.

All five destaining solutions proved to be effective at removing coffee and tea stains with respect to increasing percent light transmittance through the specimens and significantly changing their color. However, none of the destaining solutions consistently demonstrated that they were significantly better at removing both coffee and tea stains from the rough and smooth specimens with respect to both improvements in percent light transmittance through them and their color change. Of significance, Invisalign Crystals were significantly better than H_2_O_2_ at removing coffee stains from both rough and smooth specimens with respect to color change; however, since the NBS values were less than 3, the change in color was not clinically relevant. Likewise, Invisalign Crystals and Polident were significantly better than H_2_O_2_ at removing tea stains from both rough and smooth specimens with respect to color change, but, again, the NBS values were not clinically significant. H_2_O_2_ may cause the release of unpolymerized monomer and unspecific oxidative products from the polymer, which was studied in a composite polymer [27]. There were also some statistically significant differences between the cleaning solutions in their ability to remove coffee and tea stains with respect to ∆T values. However, none of the changes were greater than 1%, so these small changes in percent light transmittance were also not really clinically significant. Wible et al. also found that Essix ACE clear retainer specimens were affected by some of the same cleaning methods used in this study over a 6-month time period with respect to decreases in percent light transmittance over the time of the study; however, as in the case of this study, none of the cleaning methods were “ideal” for cleaning the copolyester material in terms of percent light transmittance [8]. 

An importing finding of this study is that the surface roughness of the retainer material plays a significant role in the ability of the cleaning solutions to remove stain. Even though the staining solutions did not significantly stain the rough and smooth specimens differently, it was shown that all of the destaining solutions had a greater effect on cleaning the rough specimens with respect to improved changes in percent light transmittance and greater changes in color. Furthermore, this effect was significant for both coffee- and tea-stained specimens, except for the ∆T values of the tea-stained rough specimens, which were greater than for the smooth specimens, although not significantly greater. This means that, when evaluating the ability of a cleaning method to remove coffee and tea stains from a retainer material, surface roughness can have a significant impact and a clinically relevant surface should be considered for the study.

Another important finding of this study is that an ultrasonic cleaning unit generally enhanced the ability of the five destaining solutions to clean tea-stained specimens with respect to both ∆T and NBS values. However, the enhancements were only significant for the ∆T values. On the other hand, the use of the ultrasonic cleaning unit did not have a significant effect on enhancing the ability of the different destaining solutions to clean coffee-stained specimens for either ∆T or NBS values. 

## 5. Conclusions

All five destaining solutions proved to be effective at removing coffee and tea stains with respect to increasing percent light transmittance of the specimens and significantly changing their color, although none of them performed significantly better at removing both types of stains on all specimens. An importing finding of this study is that the surface roughness of the retainer material plays a significant role in the ability of the cleaning solutions to remove stains with all of the destaining solutions, demonstrating a greater effect on cleaning the rough specimens with respect to improved changes in percent light transmittance and greater changes in color. This effect was significant for both coffee- and tea-stained specimens, except for the ∆T values of the tea-stained rough specimens. Additionally of importance, it was shown that an ultrasonic cleaning unit generally enhanced the ability of the five destaining solutions to clean tea-stained specimens with respect to both ∆T and NBS values. However, the enhancement was only significant for the ∆T values. Ultrasonic cleaning was not shown to have a significant effect on cleaning coffee-stained specimens.

## Figures and Tables

**Figure 1 materials-15-06808-f001:**
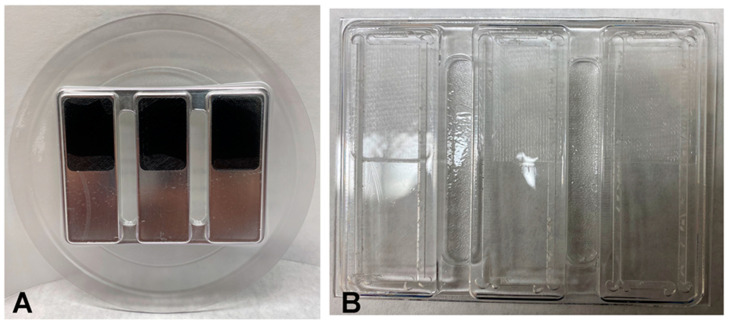
The metal molds with textured inserts (**A**). The copolyester retainer materials with different surface textures after thermoforming (**B**).

**Figure 2 materials-15-06808-f002:**
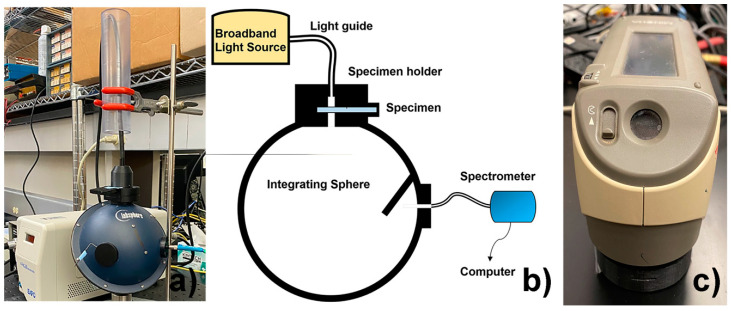
(**a**) Spectrometer/Integrating sphere system for the evaluation of percent light transmittance. (**b**) Diagram of light transmittance measurement system. (**c**) Spectrophotometer (CM-2600d Spectrophotometer, Konica Minolta, Tokyo, Japan) for color parameter change.

**Figure 3 materials-15-06808-f003:**
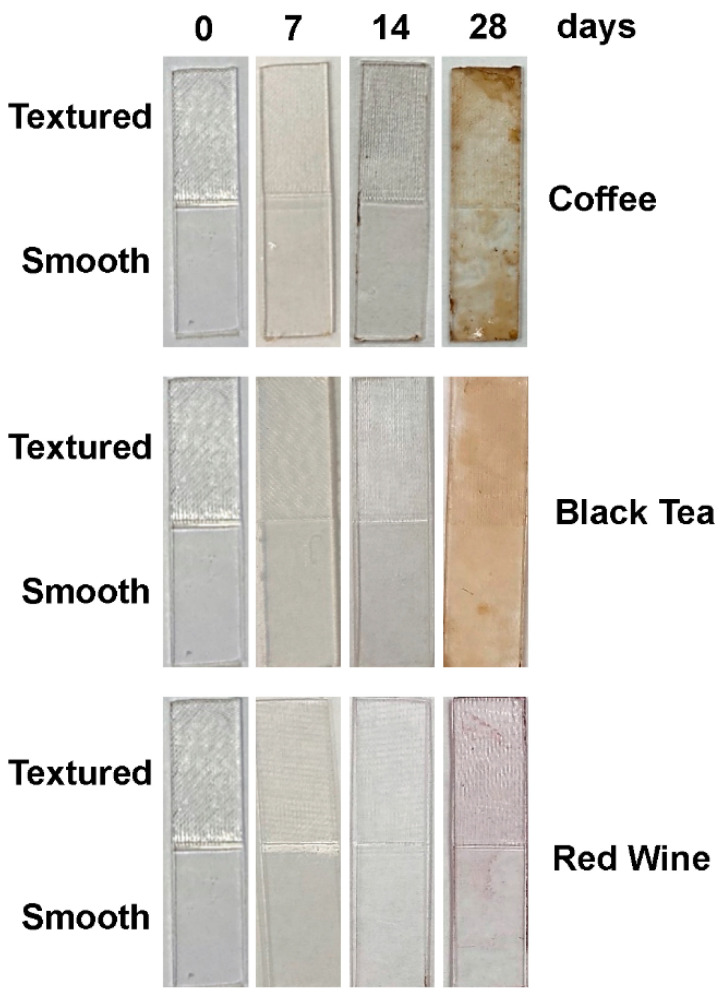
Representative examples of the copolyester retainer materials after staining by the different staining solutions at different time points.

**Table 1 materials-15-06808-t001:** The distribution of the specimens into staining groups and then, after staining, into cleaning groups.

		Four Staining Solutions							
		**Coffee** 100 specimens		**Black Tea** 100 specimens		**Red Wine** 100 specimens		**Distilled Water** 100 specimens	
		After 28 days of staining, each group of 100 specimens was divided into 5 groups of 20 for cleaning, where 10 specimens were subjected to ultrasonic agitation and 10 specimens were agitated by stirring							
**Cleaning solutions**	**Cleaning time (min)**	**Ultrasonic** 50 specimens	**Non-ultrasonic** 50 specimens	**Ultrasonic** 50 specimens	**Non-ultrasonic** 50 specimens	**Ultrasonic** 50 specimens	**Non-ultrasonic** 50 specimens	**Ultrasonic** 50 specimens	**Non-ultrasonic** 50 specimens
Invisalign^®^ Cleaning Crystals	15	10	10	10	10	10	10	10	10
Retainer Brite^®^	15	10	10	10	10	10	10	10	10
Polident^®^	3	10	10	10	10	10	10	10	10
Listerine^®^ mouthwash	15	10	10	10	10	10	10	10	10
3% hydrogen peroxide	15	10	10	10	10	10	10	10	10

**Table 2 materials-15-06808-t002:** Criteria of the National Bureau of Standards (NBS) units [19].

NBS Units	Critical Remarks of Color Differences	
0.0–0.5	Trace	Extremely slight change
0.5–1.5	Slight	Slight change
1.5–3.0	Noticeable	Perceivable
3.0–6.0	Appreciable	Marked change
6.0–12.0	Much	Extremely marked change
12.0 or more	Very much	Change to other color

Note that a value above 3 is considered of clinical significance in this study.

**Table 3 materials-15-06808-t003:** Changes in percent light transmittance, ∆T, for copolyester retainer specimens with different surface textures at different times in different staining solutions (mean ∆T value in % ± standard deviation).

	Day 7		Day 14		Day 28	
	Rough	Smooth	Rough	Smooth	Rough	Smooth
**Coffee**	1.015 ^a,b,c^ (0.441)	1.920 ^a,b,c,d^ (0.269)	1.220 (0.929)	2.101 ^b,c,d^ (0.258)	23.466 ^b,c,e,f^ (6.543)	17.429 ^b,c,e,f^ (7.931)
**Tea**	0.923 ^b,c^ (0.372)	0.957 (0.257)	1.421 (0.656)	1.190 (0.231)	25.190 ^b,c,e,f^ (2.162)	19.184 ^b,c,e,f^ (6.160)
**Wine**	−0.067 ^a^ (0.001)	0.782 ^a^ (0.162)	1.194 ^e^ (0.404)	1.051 (0.305)	2.281 ^b,e^ (1.654)	1.926 ^e,f^ (0.472)
**Water**	−1.855 (4.004)	0.494 (0.171)	0.405 (0.590)	0.697 (0.264)	0.154 (0.674)	0.812 (0.281)

^a^ significant difference between surfaces at specific day; ^b^ significant difference from water at specific day; ^c^ significant difference from wine at specific day; ^d^ significant difference from tea at specific day; ^e^ significant difference from day 7; ^f^ significant difference from day 14; ∆T = (percent light transmittance at baseline − percent light transmittance at specific day of measurement).

**Table 4 materials-15-06808-t004:** NBS values for copolyester retainer specimens with different surface textures at different times after staining in different solutions (mean NBS value ± standard deviation).

Time by Surface and Staining	Day 7		Day 14		Day 28	
	Rough	Smooth	Rough	Smooth	Rough	Smooth
**Coffee**	1.908 ^b^ (0.391)	2.615 ^b,c^ (0.613)	2.197 ^a,e^ (0.798)	3.478 ^a,b,c,d,e^ (0.634)	16.097 ^b,c,f^ (4.283)	12.793 ^b,c,f^ (5.228)
**Tea**	1.930 ^b^ (0.313)	1.750 ^b,c^ (0.072)	2.957 ^a,b,e^ (0.447)	2.250 ^a,b,c,e^ (0.110)	17.157 ^b,c,e,f^ (3.918)	14.482 ^b,c,e,f^ (5.024)
**Wine**	2.078 ^a,b^ (0.545)	1.439 ^a^ (0.129)	2.170 (0.622)	1.438 ^e^ (0.264)	3.845 ^b^ (1.535)	2.458 ^b,f^ (0.712)
**Water**	0.941 (0.354)	1.299 (0.109)	1.268 (0.446)	1.246 ^e^ (0.233)	0.925 (0.193)	0.837 ^f^ (0.171)

^a^ significant difference between surfaces at specific day; ^b^ significant difference from water at specific day; ^c^ significant difference from wine at specific day; ^d^ significant difference from tea at specific day; ^e^ significant difference from day 7; ^f^ significant difference from day 14. NBS = [(∆E* at specific day of measurement − ∆E* at baseline) × 0.92].

**Table 5 materials-15-06808-t005:** Changes in percent light transmittance, ∆T, for copolyester retainer specimens with different surface textures stained in coffee and tea solutions and then destained with five different cleaning solutions (mean ∆T value in % ± standard deviation).

Stains by Surface and Cleaning Solutions	Coffee		Tea		Wine	
	Rough	Smooth	Rough	Smooth	Rough	Smooth
**Invisalign^®^ Crystals**	22.531 ^a^ (1.094)	16.262 ^a^ (0.285)	24.108 (0.824)	21.715 (3.8665)	1.270 (0.819)	1.088 (0.584)
**Retainer Brite^®^**	22.961 ^a,d^ (0.418)	16.072 ^a^ (0.544)	24.188 (0.643)	21.452 (3.873)	2.100 ^a^ (0.931)	1.174 ^a^ (0.396)
**Listerine^®^ Mouthwash**	21.931 ^a,c^ (0.622)	15.614 ^a,b^ (0.565)	24.031 (1.941)	20.981 (4.380)	1.681 ^a^ (1.043)	0.758 ^a^ (0.534)
**Polident^®^**	22.084 ^a^ (0.837)	15.734 ^a,b^ (0.317)	23.633 (0.988)	21.902 (4.222)	1.395 ^a^ (0.587)	1.044 ^a^ (0.280)
**H_2_O_2_**	22.098 ^a^ (1.648)	15.618 ^a,b^ (0.502)	23.446 ^b,c,d^ (4.087)	20.808 (4.251)	1.576 ^a^ (0.667)	0.944 ^a^ (0.277)

^a^ significant difference between surfaces: *p* < 0.05; ^b^ significant difference between solution and Invisalign crystals: *p* < 0.05; ^c^ significant difference between solution and Retainer Brite: *p* < 0.05; ^d^ significant difference between solution and Listerine: *p* < 0.05. ∆T = (percent light transmittance after destaining − percent light transmittance before destaining).

**Table 6 materials-15-06808-t006:** NBS values for copolyester retainer specimens with different surface textures stained in coffee and tea solutions and then destained with five different cleaning solutions (mean NBS value ± standard deviation).

Stains by Surface and Cleaning Solutions	Coffee		Tea		Wine	
	Rough	Smooth	Rough	Smooth	Rough	Smooth
**Invisalign^®^ Crystals**	15.107 ^a^ (0.372)	11.013 ^a^ (0.182)	15.724 ^a^ (0.292)	13.502 ^a^ (0.161)	2.757 ^a^ (0.331)	1.729 ^a^ (0.145)
**Retainer Brite^®^**	15.024 ^a^ (0.255)	10.983 ^a^ (0.285)	15.720 ^a^ (0.289)	13.170 ^a^ (0.403)	2.701 ^a^ (0.297)	1.741 ^a^ (0.144)
**Listerine^®^ Mouthwash**	14.539 ^a^ (0.496)	10.697 ^a^ (0.468)	15.433 ^a^ (0.714)	13.175 ^a,b^ (0.284)	2.684 ^a^ (0.330)	1.782 ^a,e^ (0.174)
**Polident^®^**	14.722 ^a^ (0.343)	10.822 ^a^ (0.453)	15.731 ^a^ (0.268)	13.350 ^a^ (0.177)	2.829 ^a^ (0.336)	1.550 ^a,d^ (0.157)
**H_2_O_2_**	14.169 ^a,b,c^ (0.670)	10.580 ^a,b^ (0.251)	14.591 ^a,b,c,e^ (0.913)	12.689 ^a,b,c,e^ (0.754)	2.752 ^a^ (0.269)	1.551^a^ (0.251)

^a^ significant difference between surfaces: *p* < 0.05; ^b^ significant difference between solution and Invisalign crystals: *p* < 0.05; ^c^ significant difference between solution and Retainer Brite: *p* < 0.05; ^d^ significant difference between solution and Listerine: *p* < 0.05; ^e^ significant difference between solution and Polident: *p* < 0.05. NBS = [(∆E* before destaining − ∆E* after destaining) × 0.92].

**Table 7 materials-15-06808-t007:** Changes in percent light transmittance, ∆T, of copolyester retainer specimens stained in coffee and tea solutions and then destained with five different cleaning solutions with or without the use of an ultrasonic cleaner unit (mean ∆T value in % ± standard deviation).

Stains by Surface and Cleaning Solutions	Coffee		Tea		Wine	
	Non-Ultrasonic	Ultrasonic	Non-Ultrasonic	Ultrasonic	Non-Ultrasonic	Ultrasonic
**Invisalign^®^** **Crystals**	19.413 (3.335)	19.380 (3.462)	21.152 ^a^ (3.364)	24.675 ^a^ (0.780)	1.005 (0.874)	1.353 (0.444)
**Retainer Brite^®^**	19.511 (3.632)	19.523 (3.694)	20.950 ^a^ (3.369)	24.692 ^a^ (0.662)	2.021 ^a^ (1.030)	1.253 ^a^ (0.332)
**Listerine^®^ Mouthwash**	18.740 (3.327)	18.806 (3.436)	20.198 ^a^ (4.015)	24.8131 ^a^ (0.406)	1.625 ^a^ (1.071)	0.814 ^a^ (0.577)
**Polident^®^**	18.925 (3.161)	18.894 (3.634)	21.077 ^a^ (3.337)	24.458 ^a^ (1.726)	1.016 (0.473)	1.423 ^b^ (0.419)
**H_2_O_2_**	19.440 (3.876)	18.276 (3.243)	20.651 ^a^ (5.605)	23.603 ^a^ (1.515)	1.275 (0.802)	1.246 (0.316)

^a^ significant difference between non-ultrasonic and ultrasonic means: *p* < 0.05; ^b^ significant difference between Listerine and other solutions: *p* < 0.05. ∆T = (percent light transmittance after destaining − percent light transmittance before destaining).

**Table 8 materials-15-06808-t008:** NBS values for copolyester retainer specimens stained in coffee and tea solutions and then destained with five different cleaning solutions with or without the use of an ultrasonic cleaning cleaner unit (mean NBS value ± standard deviations).

Stains by Surface and Cleaning Solutions	Coffee		Tea		Wine	
	Non-Ultrasonic	Ultrasonic	Non-Ultrasonic	Ultrasonic	Non-Ultrasonic	Ultrasonic
**Invisalign^®^** **Crystals**	13.079 (2.222)	13.041 (2.132)	14.580 (1.286)	14.645 (1.096)	2.296 (0.623)	2.189 (0.568)
**Retainer Brite^®^**	12.832 (2.137)	13.174 (2.142)	14.279 (1.488)	14.611 (1.258)	2.150 (0.533)	2.291 (0.571)
**Listerine^®^ Mouthwash**	12.499 (2.097)	12.737 (2.059)	14.258 (1.339)	14.350 (1.275)	2.322 (0.623)	2.144 (0.432)
**Polident^®^**	12.694 (2.067)	12.911 (2.054)	14.560 (1.384)	14.522 (1.156)	2.103 (0.585)	2.277 (0.829)
**H_2_O_2_**	12.506 (2.136)	12.243 (1.752)	13.355 (1.372)	13.925 (1.612)	2.041 (0.669)	2.261 (0.680)

NBS = [(∆E* before destaining − ∆E* after destaining) × 0.92].

## Data Availability

Not applicable.

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
