# Peer review of "The Effects of Staining and Cleaning on the Color and Light Transmittance Changes of a Copolyester Retainer Material with Different Surface Textures"

_materials, 2022, doi:10.3390/ma15196808_

Round 1

Reviewer 1 Report

Staining and Cleaning Effects on Color and Light Transmittance Changes of a Copolyester Retainer Material with Different Surface Textures.

Laurie Susarchick, Insia Virji, Grace Viana, Mervat Mahmoud, Veerasathpurush Allareddy, Max Gruber, Henry Lukic, Spiro Megremis, Phimon Atsawasuwan, *

Recommendation: Major Revision

The manuscript by Susarchick titled “Staining and Cleaning Effects on Color and Light Transmittance Changes of a Copolyester Retainer Material with Different Surface Textures.” The authors provided a detailed experimental procedure and discussion of the results well-articulated. However, this paper requires a major revision before it can be accepted for publication.

The rationale of this study should be included accordingly. What makes this study different from the methods reported in literature?

The manuscript can be published after a major revision.

The following corrections should be made:

1.      Line 79: Figure 2. The distribution of specimens into staining groups and, then after staining, into cleaning groups. Should be changed to Table 1. The distribution of specimens into staining groups and, then after staining, into cleaning groups. Consequently, the authors should renumber the tables accordingly.

2.      Line 103:  Absolute percent light transmittance was determined based on a previously published method for measuring the translucency of dental ceramics.

Why was this method chosen in this study? Did the authors carry out any improvement on the method?

3.      Line 122: Figure 3. Label the Figures (a), (b) and (c) on the diagram of Figure 3)

4.      Line 167: This is also shown qualitatively in Figure 5. Should be corrected to Table 5

5.      Line 226-228: However, the only statistically significant differences were for the rough specimens, with the H2O2 solution being significantly less effective than the other solutions except for the Polident solution. Can authors explain why Polident solution exhibited an effective result in comparison to other solutions?

6.      Line 293-294: Providers should clearly inform patients to remove retainers before eating and drinking, as staining could occur from food or drinks that would allow the stain to accumulate on the retainer. I really like this finding? Can the authors provide any suggestion on regulation policy that can back up this claim?  

Reviewer 2 Report

Right from the beginning of your contribution a very clearly laid out protocol became evident. This is very much appreciated.

Apart from 4 commercially available destaining products, hydrogen peroxide was tested. Since the authors declare no conflict of interest it is assumed that no of the producers sponsored this study in any way.

The ingredients of the commercial products are not mentioned, so it would help to classify them a bit further. Note that e.g. the product Invisalign® Cleaning Crystals might be addressing more the sanitizing rather the bleaching effect.

Your introduction helps also readers being not familiar with the study subject to understand the needed basics. Given just the numbers of patients treated, it is surprising that apparently so far only one group (and one journal) , Ref. [6-8], has addressed this issue.

From line 207 I understood that the wine stained samples will not be evaluated further, however, in the following (like Table 4) these samples are still included. Maybe rephrase carefully.

Finally, you concluded that the application of the tested commercial products is effective and even cheaper (according to my survey) for the consumer then applying H2O2 solutions.

Science, text, figures: All good!

Round 2

Reviewer 1 Report

The authors have addressed my concerns.

Reviewer 2 Report

Corrections made are ok.